# Dissecting Submission Limit in Desk-Rejections: A Mathematical Analysis of Fairness in AI Conference Policies

**Yuefan Cao** [1]  **Xiaoyu Li** [2]  **Yingyu Liang** [3][4]  **Zhizhou Sha** [5]  **Zhenmei Shi** [4]  **Zhao Song** [6]  **Jiahao Zhang** [1]

## Abstract

As AI research surges in both impact and volume, conferences have imposed submission limits to maintain paper quality and alleviate organizational pressure. In this work, we examine the fairness of desk-rejection systems under submission limits and reveal that existing practices can result in substantial inequities. Specifically, we formally define the paper submission limit problem and identify a critical dilemma: when the number of authors exceeds three, it becomes impossible to reject papers solely based on excessive submissions without negatively impacting innocent authors. Thus, this issue may unfairly affect early-career researchers, as their submissions may be penalized due to co-authors with significantly higher submission counts, while senior researchers with numerous papers face minimal consequences. To address this, we propose an optimization-based fairness-aware desk-rejection mechanism and formally define two fairness metrics: worst-case fairness and average fairness. We prove that optimizing worst-case fairness is NP-hard, whereas average fairness can be efficiently optimized via linear programming. Through case studies, we demonstrate that our proposed system ensures greater equity than existing methods, including those used in CVPR 2025, offering a more socially just approach to managing excessive submissions in AI conferences.

---
[1]University of Washington  [2]University of New South Wales  [3]The University of Hong Kong  [4]University of Wisconsin–Madison  [5]University of Texas at Austin  [6]University of California, Berkeley. Correspondence to: Xiaoyu Li <7.xiaoyu.li@gmail.com>, Yingyu Liang <yingyul@hku.hk, yliang@cs.wisc.edu>, Zhenmei Shi <zhmeishi@cs.wisc.edu>, Zhao Song <magic.linuxkde@gmail.com>.

*Proceedings of the $42^{nd}$ International Conference on Machine Learning*, Vancouver, Canada. PMLR 267, 2025. Copyright 2025 by the author(s).

## 1. Introduction

We are living in an era shaped by the unprecedented advancements of Artificial Intelligence (AI), where transformative breakthroughs have emerged across various domains in just a few years. A key driving force behind AI's rapid progress is the prevalence of top conferences held frequently throughout the year, offering dynamic platforms to present many of the field's most influential papers. For example, ResNet (He et al., 2016), a foundational milestone in deep learning with over 250,000 citations, was first introduced at CVPR 2016. Similarly, the Transformer architecture (Vaswani et al., 2017), the backbone of modern large language models, emerged at NeurIPS 2017. More recently, diffusion models (Ho et al., 2020), which represent the state-of-the-art in image generation, were presented at NeurIPS 2020, while CLIP (Radford et al., 2021), a leading model for image-text pretraining, was showcased at ICML 2021. These groundbreaking contributions from top-tier conferences have significantly accelerated the advancement of AI, enriching both theoretical insights and practical applications.

As AI continues to expand its applications and capabilities in real-world domains such as dialogue systems (Schulman et al., 2022; Achiam et al., 2023; Anthropic, 2024), image generation (Ho et al., 2020; Song et al., 2021), and video generation (Ho et al., 2022; Blattmann et al., 2023), its immense potential for commercialization has raised growing enthusiasm in AI research. This enthusiasm has led to a rapid, rocket-like increase in the number of AI-related papers in 2024, as witnessed by recent studies (Stanford, 2024). A direct consequence of this surge is the significant rise in submissions to AI conferences, which has placed a heavy burden on program committees tasked with selecting papers for acceptance. To address these challenges and maintain the quality of accepted papers, many leading conferences have introduced submission limits per author. In 2025, a wide range of leading AI conferences, including CVPR, ICCV, AAAI, WSDM, IJCAI, and KDD, introduced submission limits per author in their guidelines, ranging from a maximum of $x = 7$ to $x = 25$. Table 1 provides an overview of these submission limits across major AI conferences.

However, such a desk-rejection mechanism may result in un-

*Table 1.* In this table, we summarize the submission limits of top conferences in recent years. For details of each conference website, we refer the readers to Section D in the Appendix. Some conferences (CVPR, ICCV, WSDM, KDD) employ a conventional desk-reject algorithm (Algorithm 2), where papers are desk-rejected once an author has registered more than $x$ (the submission limit) papers. We denote the absence of a limit with "N/A".

| Conference Name | Year | Submission Limit |
| --- | --- | --- |
| CVPR | 2025 | 25 |
| CVPR | 2024 | N/A |
| ICCV | 2025 | 25 |
| ICCV | 2023 | N/A |
| AAAI | 2023-2025 | 10 |
| AAAI | 2022 | N/A |
| WSDM | 2021-2025 | 10 |
| WSDM | 2020 | N/A |
| IJCAI | 2021-2025 | 8 |
| IJCAI | 2020 | 6 |
| IJCAI | 2018-2019 | 10 |
| IJCAI | 2017 | N/A |
| KDD | 2024-2025 | 7 |
| KDD | 2023 | N/A |

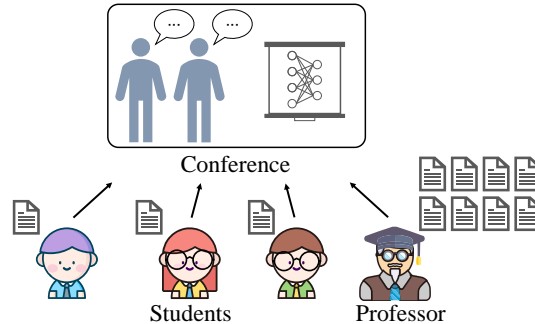

*Figure 1.* The Matthew Effect in the AI community. This figure illustrates the worsening Matthew Effect in the AI community, where senior researchers tend to have a significantly higher number of submissions, while junior researchers have relatively few.

intended negative societal impacts due to the Matthew effect in the research community (Bol et al., 2018), as illustrated in Figure 1. Recent research has shown that the impact of a setback (e.g., a paper rejection) is often much greater for early-career researchers than for senior researchers (Wang et al., 2019; Sun et al., 2023), which shows that the effect of a desk rejection can vary significantly depending on the author's career stage. For instance, as illustrated in Figure 2, consider the case of a young student submitting their only draft to the conference, co-authored with a renowned researcher who submits numerous papers annually. If the paper is desk-rejected due to exceeding submission limits, the senior researcher might view this as a neglectable inconvenience. In contrast, the rejection could have severe consequences for the student, as the paper might be crucial for applying to graduate programs, securing employment, or forming a chapter of their thesis. This disparity in the impact of desk rejections may worsen the Matthew effect in the AI community by disproportionately disadvantaging researchers with only one or two submitted papers, while having little effect on prolific senior researchers. Such outcomes raise important concerns about fairness and equity in current desk-rejection policies.

In response to the challenges posed by paper-limit-based desk-rejection systems, this work investigates an important and practical problem: ensuring fairness in desk-rejection systems for AI conferences under submission limits. As illustrated in Figure 3, our goal is to design a fair desk-rejection system that prioritizes rejecting submissions from authors with many papers while protecting those with fewer submissions, particularly early-career researchers. Our key contributions are as follows:

- We formally define the paper submission limit problem in desk-rejection systems and prove that an ideal system that rejects papers solely based on each author's excessive submissions is mathematically impossible when there are more than three authors.

- We introduce two fairness metrics: worst-case fairness and average fairness. We formulate the fairness-aware paper submission limit problem as an integer programming problem. We formally prove that optimizing worst-case fairness is NP-hard, while the average fairness optimization problem can be solved efficiently using any off-the-shelf linear programming solver.

- Through case studies, we demonstrate that our proposed system achieves greater fairness compared to existing approaches used in top AI conferences such as CVPR 2025, promoting social justice and fostering a more inclusive ML research community.

**Roadmap.** Our paper is organized as follows: In Section 2, we review related literature. In Section 3, we present the key definition of the paper submission limit problem. In Section 4, we show that no algorithm can reach the ideal desk-rejection system without unfair collective punishments. In Section 5, we present our new fairness-aware desk-rejection system. In Section 6, we show by case studies that our system is better than existing systems. In Section 8, we present our conclusions and discuss future directions.

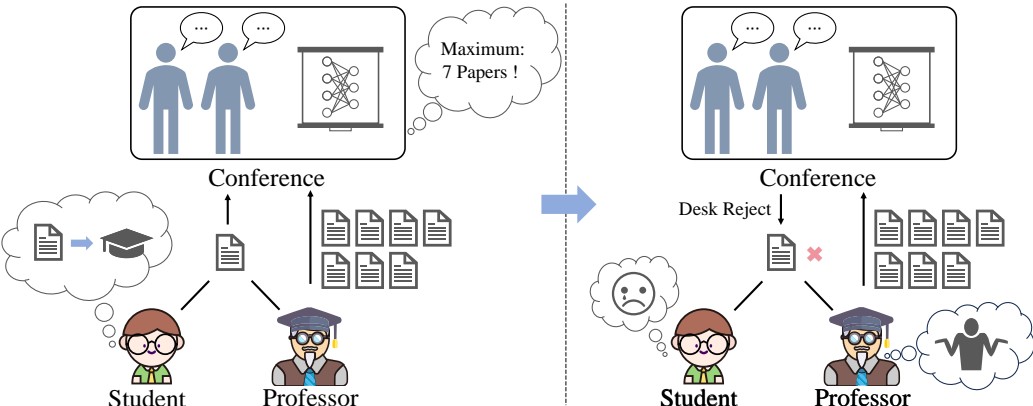

*Figure 2.* The unfairness of desk rejection based on submission limits. **Left: A careless mistake.** In this scenario, a young student submits the only paper, co-authored with a professor who submits numerous papers, and carelessly exceeds the submission limit. The paper, which may aim to apply to graduate programs, secure employment, or form a chapter of the thesis, is very important for the student but may not be for the professor. **Right: The desk rejection.** If the paper is desk-rejected due to submission limits, it poses a minor inconvenience to the professor, and the professor can shrug about it due to his remaining papers. However, it could have severe consequences for the students, as the paper is crucial for the student's future plans.

## 2. Related Works

### 2.1. Desk Reject Mechanism

A wide range of desk-rejection mechanisms have been developed to reduce the human effort involved in the peer-review process (Ansell & Samuels, 2021). One of the most widely adopted desk-rejection rules is rejecting papers that violate anonymity requirements (Jefferson et al., 2002; Tennant, 2018). This rule is crucial for maintaining unbiased evaluations of researchers from diverse institutions and career levels while preventing conflicts of interest. Another common mechanism addresses duplicate and dual submissions (Stone, 2003; Leopold, 2013), alleviating the duplication of reviewer efforts across multiple venues and upholding ethical publication standards. Additionally, plagiarism (King & ChatGPT, 2023; Elali & Rachid, 2023) is a major concern in desk rejections at AI conferences, as it undermines the integrity of the academic community, violates intellectual property rights, and compromises the originality and credibility of research. In response to the growing number of submissions to AI conferences, new types of desk-rejection rules have recently emerged (Leyton-Brown et al., 2024). For example, IJCAI 2020 and NeurIPS 2020 implemented a fast desk-rejection mechanism, allowing area chairs to reject papers based on a quick review of the abstract and main content to manage the review workload. However, this approach introduced noise and sometimes resulted in the rejection of generally good papers, leading to its reduced prevalence compared to more systematic mechanisms like enforcing submission limits, which is the main focus of this paper. To the best of our knowledge, limited literature has explored these emerging desk-rejection techniques, and our work is among the first to formally study the desk-rejection

mechanism based on maximum submission limits.

### 2.2. The Competitive Race in AI Publication

Due to the rapid increase in submissions to AI conferences in recent years (Stanford, 2024), concerns about the intense competition in these conferences are growing. As Bengio Yoshua noted (Bengio, 2020): "It is more competitive, everything is happening fast and putting a lot of pressure on everyone. The field has grown exponentially in size. Students are more protective of their ideas and in a hurry to put them out, by fear that someone else would be working on the same thing elsewhere, and in general, a PhD ends up with at least 50% more papers than what I gather it was 20 or 30 years ago." Consequently, paper acceptance has become increasingly critical in AI job applications (Ahmed, 2022; Besiroglu et al., 2024), as having more papers is now the norm. Therefore, it is crucial to establish fair and practical guidelines for desk rejections (Teixeira da Silva et al., 2018), ensuring that every group of authors is treated equitably in AI conferences.

### 2.3. Fairness System Design

Fairness (Francez, 2012; Mehrabi et al., 2021) is a key principle of social justice, reflecting the absence of bias toward individuals or groups based on inherent characteristics. Due to its profound societal impact, fairness has become an essential consideration in the design of algorithms across various computer systems that interact with human factors. In recommender systems, fairness can manifest in various forms, such as item fairness (Zhang et al., 2021; Ge et al., 2021), which ensures that items from different categories or with varying levels of prior exposure are recommended equi-

tably, and user fairness (Li et al., 2021a;b), which guarantees that all users, regardless of their backgrounds or preferences, have equal opportunities to access relevant content. These fairness measures help balance opportunities for both users and retailers, fostering equity in the recommendation process. In candidate selection systems (Gilliland, 1993; Wang et al., 2020), fairness ensures that all candidates are evaluated solely on merit, independent of factors such as race, gender, or socio-economic background, promoting equality and ensuring that the selection processes are inclusive. In information access systems (Ekstrand et al., 2022), including job search (Wu et al., 2022), ranking (Yang et al., 2023), music discovery (Melchiorre et al., 2021), and selection problems (Emelianov et al., 2020), fairness guarantees that all individuals can access the information they need without discrimination, ensuring equal opportunities for users to make informed decisions. Similarly, in dialog systems (Guo et al., 2022; Gallegos et al., 2024), fairness ensures that language models avoid generating biased text or making inappropriate word-context associations related to social groups, supporting equitable and respectful interactions. Moreover, recent research has investigated group fairness in peer review processes for AI conferences, highlighting the importance of equitable evaluation for submissions (Aziz et al., 2023). Despite the widespread focus on fairness in algorithmic design, the fairness of desk-rejection mechanisms remains an open question and serves as the primary motivation for this paper.

## 3. Preliminary

In this section, we first introduce the notations in Section 3.1. Then, we present the general problem formulation in Section 3.2.

### 3.1. Notations

For any positive integer $n$, we use $[n]$ to denote the set $\{1, 2, \ldots, n\}$. We use $\mathbb{N}_+$ to represent the set of all positive integers. For two sets $\mathcal{B}$ and $\mathcal{C}$, we denote the set difference as $\mathcal{B} \setminus \mathcal{C} := \{x \in \mathcal{B} : x \notin \mathcal{C}\}$. For a vector $x \in \mathbb{R}^d$, $\mathrm{Diag}(d)$ denotes a diagonal matrix $X \in \mathbb{R}^{d \times d}$, where the diagonal entries satisfy $X_{i,i} = x_i$ for all $i \in [d]$, and all off-diagonal entries are zero. We use $\mathbf{1}_n$ to denote an $n$-dimensional column vector with all entries equal to one.

### 3.2. Problem Formulation

In this section, we further introduce the actual problem we will investigate in this paper, where we begin with introducing the definition for three kinds of authors that will appear later in our discussion.

**Definition 3.1** (Submission Limit Problem). *Let $\mathcal{A} = \{a_1, a_2, \ldots, a_n\}$ denote the set of $n$ authors, and let $\mathcal{P} =*

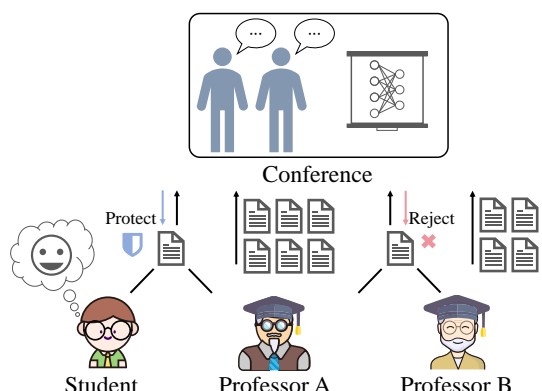

*Figure 3.* Our research objective. This figure presents the goal of our study: creating a more equitable desk-rejection system. Consider Professor A, who has carelessly submitted numerous papers exceeding the submission limit, collaborating with another senior researcher (Professor B) with many submissions, and a young student with only one paper. Our proposed system prioritizes desk-rejecting papers from authors with a large number of submissions first, thereby increasing the student's chances of having their paper accepted. This approach aims to mitigate the disparity in the impact of desk rejections and promote fairness.

$\{p_1, p_2, \ldots, p_m\}$ *denote the set of $m$ papers. Each author $a_i \in \mathcal{A}$ has a subset of papers $P_i \subseteq \mathcal{P}$, and each paper $p_j \in \mathcal{P}$ is authored by a subset of authors $A_j \subseteq \mathcal{A}$. For each author, $a_i \in \mathcal{A}$, let $C_i$ denote the set of all coauthors of $a_i$ and let $x \in \mathbb{N}_+$ denote the maximum number of papers each author can submit.*

*The goal is to find a subset $S \subseteq \mathcal{P}$ of papers (to keep) such that for every $a_i \in \mathcal{A}$,*

$$\underbrace{|\{p_j \in S : a_i \in A_j\}|}_{\text{\#remained papers of author } a_i} \leq x.$$

*or equivalently find a subset $\overline{S} \subseteq \mathcal{P}$ of papers (to reject) such that for every $a_i \in \mathcal{A}$,*

$$|P_i| - \underbrace{|\{j \in \overline{S} : i \in A_j\}|}_{\text{\#rejected papers of author } a_i} \leq x.$$

We now present several fundamental facts related to Definition 3.1, which can be easily verified through basic set theory.

**Fact 3.2.** *For any author $a_i \in \mathcal{A}$ and paper $p_j \in \mathcal{P}$, $a_i \in A_j$ if and only if $p_j \in P_i$.*

**Fact 3.3.** *For each author $a_i \in \mathcal{A}$, the number of papers submitted by the author can be formulated as:*

$$|P_i| = |\{p_j \in \mathcal{P} : a_i \in A_j\}|.$$

**Fact 3.4.** *For each paper $j \in [m]$, the number of authors of this paper can be formulated as: $|A_j| = |\{a_i \in \mathcal{A} : p_j \in P_i\}|.$*

**Fact 3.5.** *For each author $a_i \in \mathcal{A}$, the set of coauthors for author $a_i$ can be formulated as:* $C_i = (\bigcup_{p_j \in P_i} A_j) \setminus \{a_i\}$.

# 4. The Desk Rejection Dilemma

In this section, we define the concept of an ideal desk-rejection system in Section 4.1 and formally demonstrate in Section 4.2 that no algorithm can achieve this ideal system.

## 4.1. Ideal Desk-Rejection

An ideal desk-rejection system should avoid unfairly rejecting papers from authors who either comply with the submission limit or exceed it by only one or two papers. Otherwise, authors may face consequences due to co-authors with an excessively high number of submissions. This issue is particularly problematic for early-career researchers, as such collective penalties can have a significant negative impact on their careers.

To address this, we formally define the criteria for an ideal desk-rejection outcome for the problem in Definition 3.1, where rejections are based solely on an author's excessive submissions, without unfairly penalizing others.

**Definition 4.1** (Ideal desk-rejection). *An ideal solution for the submission limit problem in Definition 3.1 is a paper subset $S \subseteq \mathcal{P}$ such that every author has exactly $\min\{x, |P_i|\}$ papers remaining after desk rejection.*

**Remark 4.2.** *The ideal desk-rejection in Definition 4.1 ensures that innocent authors with less than $x$ submissions will retain all their papers, and a non-compliant author $a_i$ with more than $x$ submissions will be desk-rejected exact $(|P_i| - x)$ papers.*

Thus, if there exists an algorithm that can reach the aforementioned ideal solution, we can ensure that no author is unfairly penalized due to their co-authors' submission behavior, achieving both fairness and individual accountability.

## 4.2. Hardness of Ideal Desk-Rejection

Unfortunately, we find that achieving an ideal desk-rejection system is fundamentally intractable. The main result regarding this hardness is presented in the following theorem:

**Theorem 4.3** (Hardness of Ideal Desk-Rejection). *Let $n = |\mathcal{A}|$ denote the number of authors in Definition 3.1. We can show that*

- **Part 1:** *For $n \leq 2$, there always exists an algorithm that can achieve the ideal desk-rejection in Definition 4.1.*

- **Part 2:** *For $n \geq 3$, there exists at least one problem instance where no algorithm can guarantee achieving*

*the ideal desk-rejection in Definition 4.1.*

*Proof.* For **Part 1**, the result follows directly from Lemma A.3 and Lemma A.5. For **Part 2**, the result is established using Lemma A.6 and Lemma A.7. Detailed technical proofs for these lemmas are provided in Appendix A. □

Therefore, since an ideal desk-rejection system is not achievable, it is inevitable that some authors may face excessive desk-rejections due to collective punishments. This challenge is particularly concerning for early-career researchers with only one or two submissions, motivating the need to seek an approximate solution that optimizes fairness in desk-rejection systems.

# 5. Fairness-Aware Desk-Rejection

In this section, we first introduce two fairness metrics in Section 5.1, and then present the hardness result on minimizing one of them in Section 4.2. In Section 5.3, we show our optimization-based fairness-aware desk-rejection framework.

## 5.1. Fairness Metrics

As discussed earlier, achieving an ideal desk-rejection system is practically infeasible, as unintended rejections due to collective punishments are unavoidable. To address this, we relax the ideal system into an approximate form, where some unfair desk-rejections are permitted, while these rejections should be proportional to each author's total number of submissions.

Specifically, we introduce a cost function for each author, which estimates the impact of desk-rejection on each author:

**Definition 5.1** (Cost Function). *Considering the submission limit problem in Definition 3.1, we define the cost function $c : [n] \times 2^{[m]} \to [0, 1]$ for a specific author $a_i$ and a set of remaining paper $S$ as*

$$c(a_i, S) := \frac{|P_i| - |\{p_j \in S : a_i \in A_j\}|}{|P_i|}.$$

**Remark 5.2.** *The cost function $c(a_i, S)$ measures the proportion of papers authored by $a_i$ that are rejected, prioritizing fairness for early-career authors with fewer submissions and aiming to reduce setbacks for them.*

To further demonstrate how this author-wise cost function could benefit fairness, we present the following example:

**Example 5.3.** *Consider a submission limit problem with $x = 10$ and $n = 2$. Suppose author $a_1$ submits papers $p_1, p_2, \ldots, p_{11}$, and author $a_2$ submits only paper $p_{11}$. Rejecting paper $p_{11}$ (i.e., $S = \mathcal{P} \setminus \{p_{11}\}$) results in a cost of $c(a_1, S) = 1/11$ for $a_1$ but a cost of $c(a_2, S) = 1$ for*

$a_2$, which is unfair to $a_2$. On the other hand, if we reject paper $p_1$ (i.e., $S' = \mathcal{P} \setminus \{p_1\}$), the cost for $a_1$ remains $c(a_1, S') = 1/11$, while the cost for $a_2$ becomes $c(a_2, S') = 0$. This minimizes both the highest cost and the total cost. This example demonstrates that our cost function encourages rejecting papers from authors with many submissions while protecting authors with few submissions.

To ensure fair treatment for all authors and avoid imposing excessive setbacks on early-career researchers, we introduce two fairness metrics based on our cost function. These metrics are inspired by the principles of utilitarian social welfare and egalitarian social welfare (Aziz et al., 2024). We begin by defining worst-case fairness, which is a strict worst-case fairness metric that aligns with the egalitarian social welfare framework by estimating the individual cost among all authors.

**Definition 5.4** (Worst-case Fairness). *Let $c : [n] \times 2^{[m]} \to [0, 1]$ be the cost function defined in Definition 5.1. We define function $\zeta_{\mathrm{worst}} : 2^{[m]} \to [0, 1]$ to measure the worst-case fairness:*

$$\zeta_{\mathrm{worst}}(S) := \max_{i \in [n]} c(a_i, S).$$

Next, we present the concept of average fairness, which aligns with utilitarian social welfare and measures the total cost across all authors.

**Definition 5.5** (Average Fairness). *Let $c : [n] \times 2^{[m]} \to [0, 1]$ be the cost function defined in Definition 5.1. We define function $\zeta_{\mathrm{avg}} : 2^{[m]} \to [0, 1]$ to measure the average fairness:*

$$\zeta_{\mathrm{avg}}(S) := \frac{1}{n} \sum_{i \in [n]} c(a_i, S).$$

To show the relationship between these two fairness metrics, we have the following proposition:

**Proposition 5.6** (Relationship of Fairness Metrics, informal version of Proposition B.1 in Appendix B). *For any solution $S \subseteq \mathcal{P}$ to the submission limit problem in Definition 3.1, we have $\zeta_{\mathrm{avg}}(S) \leq \zeta_{\mathrm{worst}}(S)$.*

## 5.2. Hardness of Worst-case Fairness-Aware Submission Limit Problem

After presenting fairness metrics for the desk-rejection system, we introduce an optimization-based framework to address these metrics. We first study the worst-case fairness-aware submission limit problem to minimize the worst-case fairness measure $\zeta_{\mathrm{worst}}$ in Definition 5.4.

**Definition 5.7** (Worst-case Fairness-Aware Submission Limit Problem). *We consider the following optimization problem:*

$$\min_{S \subseteq \mathcal{P}} \zeta_{\mathrm{worst}}(S)$$
$$\text{s.t. } |\{p_j \in S : a_i \in A_j\}| \leq x, \quad \forall a_i \in \mathcal{A}.$$

To represent the fairness metric minimization problem in matrix form, we introduce the following definition:

**Definition 5.8** (Author-Paper Matrix). *Let $W \in \{0, 1\}^{n \times m}$ denote the author-paper matrix for the author set $\mathcal{A}$ and paper set $\mathcal{P}$. Then, we define $W_{i,j} = 1$ if author $a_i$ is a coauthor of paper $p_j$, and $W_{i,j} = 0$ otherwise.*

Therefore, we present a more tractable integer programming form of the original problem and prove its equivalence to the original formulation:

**Definition 5.9** (Worst-Case Fairness-Aware Submission Limit Problem, Matrix Form). *We consider the following integer optimization problem:*

$$\min_{r \in \{0,1\}^m} \|\mathbf{1}_n - D^{-1} W r\|_\infty$$
$$\text{s.t. } (Wr)/x \leq \mathbf{1}_n$$

*where $D = \mathrm{Diag}(|P_1|, \cdots, |P_n|)$, and the rejection vector $r \in \{0, 1\}^m$ is a 0-1 vector, with $r_j = 1$ indicating that paper $p_j$ is remained, and $r_j = 0$ indicating that it is desk-rejected.*

**Proposition 5.10** (Matrix Form Equivalence for $\zeta_{\mathrm{worst}}$, informal version of Proposition B.3 in Appendix B). *The worst-case fairness-aware submission limit problem in Definition 5.7 and the matrix form integer programming problem in Definition 5.9 are equivalent.*

Unfortunately, solving this integer programming problem is highly non-trivial, which means it may not yield a feasible solution within a reasonable time for large-scale conference submission systems. We establish the computational hardness of this problem in the following theorem:

**Theorem 5.11** (Hardness, informal version of Theorem B.7 in Appendix B.2). *The Worst-Case Fairness-Aware Submission Limit Problem defined in Definition 5.7 is NP-hard.*

Since minimizing worst-case fairness is computationally intractable, our fairness-aware desk-rejection system instead focuses on minimizing average fairness.

## 5.3. Average Fairness Optimization

Given the inherent hardness of worst-case fairness optimization, we address the fairness problem using an alternative yet equally important metric: average fairness, as defined in Definition 5.5. This metric is not only a crucial fairness measure in its own right but also serves as a lower bound for

worst-case fairness as stated in Proposition 5.6, potentially improving worst-case fairness implicitly.

Following a similar approach in Section 5.2, we first formulate the submission limit problem with respect to average fairness and derive a more tractable integer programming formulation in matrix form:

**Definition 5.12** (Average Fairness-Aware Submission Limit Problem). *We consider the following optimization problem:*

$$\min_{S \subseteq \mathcal{P}} \zeta_{\text{avg}}(S)$$
$$\text{s.t. } |\{p_j \in S : a_i \in A_j\}| \le x, \quad \forall a_i \in \mathcal{A}.$$

**Definition 5.13** (Average Fairness-Aware Submission Limit Problem, Matrix Form). *We consider the following integer programming problem:*

$$\max_{r \in \{0,1\}^m} \mathbf{1}_n^\top D^{-1} W r$$
$$\text{s.t. } (Wr)/x \le \mathbf{1}_n,$$

*where $D = \text{Diag}(|P_1|, \cdots, |P_n|)$, and the rejection vector $r \in \{0,1\}^m$ is a 0-1 vector, with $r_j = 1$ indicating that paper $p_j$ is remained, and $r_j = 0$ indicating that it is desk-rejected.*

**Proposition 5.14** (Matrix Form Equivalence for $\zeta_{\text{avg}}$, informal version of Proposition B.8 in Appendix B). *The fairness-aware submission limit problem in Definition 5.12 and the matrix form integer programming problem in Definition 5.13 are equivalent.*

However, solving integer programming problems is practically challenging. To this end, we first relax the feasible region of $r$ to $[0,1]^m$, and then analyze the resulting relaxed problem.

**Definition 5.15** (Average Fairness-Aware Submission Limit Problem, Relaxation). *We consider the optimization problem*

$$\max_{r \in [0,1]^m} \mathbf{1}_n^\top D^{-1} W r$$
$$\text{s.t. } (Wr)/x \le \mathbf{1}_n,$$

*where $D = \text{Diag}(|P_1|, \cdots, |P_n|)$, and the rejection vector $r \in \{0,1\}^m$ is a 0-1 vector, with $r_j = 1$ indicating that paper $p_j$ is remained, and $r_j = 0$ indicating that it is desk-rejected.*

Fortunately, the relaxed problem is a linear program, which can be efficiently solved using standard linear programming solvers. Moreover, its optimal solution is equivalent to that of the original integer programming problem, an this result is formalized in the following theorem:

**Theorem 5.16** (Optimal Solution Equivalence of the Relaxed Problem, informal version of Theorem B.9 in Appendix B). *The optimal solution of the relaxed linear programming problem in Definition 5.15 is equivalent to the*

---

**Algorithm 1** Fairness-Aware Desk-Reject Algorithm

1: /* $\mathcal{A}$ denotes the set of $n$ authors. */
2: /* $\mathcal{P}$ denote the set of $m$ papers. */
3: /* Author $a_i \in \mathcal{N}$ has a subset of papers $P_i \subset \mathcal{P}$. */
4: /* Paper $p_j \in \mathcal{P}$ is coauthored by a subset of authors $A_j \subseteq \mathcal{A}$.*/
5: /* $x$ represents the submission limit for each author.*/
6: **procedure** FAIRDESKREJECT($\mathcal{A}, \mathcal{P}, x$)
7:     /* Initialize the constants of the problem. */
8:     **for** $i \in [n], j \in [m]$ **do**
9:         **if** $p_j \in \mathcal{A}_i$ **then**
10:             $W_{i,j} \leftarrow 1$
11:         **else**
12:             $W_{i,j} \leftarrow 0$
13:         **end if**
14:     **end for**
15:     $D \leftarrow \text{Diag}(|P_1|, \ldots, |P_n|)$
16:     /* Solve the linear programming problem in Definition 5.15. */
17:     $r^\star \leftarrow \text{LPSolver}(W, D, x, r^0)$
18:     /* Transform the solution. */
19:     $S \leftarrow \emptyset$
20:     **for** $j \in [m]$ **do**
21:         **if** $r_j = 1$ **then**
22:             $S \leftarrow S \cup \{p_j\}$
23:         **end if**
24:     **end for**
25:     **return** $S$
26: **end procedure**

---

*optimal solution of the original integer programming problem in Definition 5.13.*

This theoretical result is significant as we formally establish that the average fairness-aware submission problem in Definition 5.12 reduces to a linear programming (LP) problem with guaranteed optimality, solvable using off-the-shelf LP solvers. We formalize this procedure in Algorithm 1, where LPSolver denotes any standard LP solver, including but not limited to the simplex method (Dantzig, 1951), ellipsoid method (Khachiyan, 1980), interior-point method (Karmarkar, 1984; Renegar, 1988; Lee & Sidford, 2014; Cohen et al., 2019; Song, 2019; Brand et al., 2020; Song & Yu, 2021; Jiang et al., 2021; Cohen et al., 2021; Gu & Song, 2022; Qin et al., 2023; Liu et al., 2023; Gu et al., 2025).

**Remark 5.17.** *The time complexity of our fairness-aware desk-rejection algorithm in Algorithm 1 aligns with modern linear programming solvers. For instance, using the stochastic central path method (Cohen et al., 2021; Jiang et al., 2021; Qin et al., 2023), it achieves a time complexity of $O^*(m^{2.37} \log(m/\delta))$, where $\delta$ represents the relative accuracy corresponding to a $(1 + \delta)$-approximation guarantee.*

**Remark 5.18.** *In practice, major AI conferences routinely process submissions at the scale of $m \sim 10^4$ (Stanford, 2024). Given this regime, our algorithm guarantees efficient computation, enabling fairness-aware desk rejection within tractable timeframes, even for large-scale conferences.*

## 6. Case Study

Since desk-rejection data from top AI conferences is not publicly available, and fully open-review conferences like ICLR do not impose submission limits, evaluating real-world conference submissions is impractical. Therefore, we present a case study to demonstrate how our proposed desk-rejection algorithm more effectively addresses fairness issues. Additional case studies are provided in Appendix C.

Let the paper subscript $j$ in $p_j \in \mathcal{P}$ denote the submission order. We analyze the widely used desk-rejection system (e.g., CVPR 2025) in Algorithm 2, which rejects all papers submitted after an author's $x$-th submission. To highlight its limitations, we present a minimal working example:

**Example 6.1.** *Consider a submission limit problem as defined in Definition 3.1 with $n = 2$, $x = 25$, and $m = 26$. Author $a_1$ submits all papers $p_1, \ldots, p_{26}$, while author $a_2$ submits only $p_{26}$.*

Given the ideal desk-rejection criteria in Definition 4.1, it is evident that we can reject any papers in $\{p_1, p_2, \ldots, p_{25}\}$ following the techniques in Lemma A.4. After rejection, since $a_1$ retains 25 papers and $a_2$ retains 1 paper, the fairness metrics are $\zeta_{\text{worst}}(S) = \max\{1/26, 0\} = 1/26$ and $\zeta_{\text{avg}}(S) = \frac{1}{2}(1/26 + 0) = 1/52$.

On the other hand, the CVPR 2025 algorithm, as described in Algorithm 2, rejects $p_{26}$, retaining $S = \{p_1, \ldots, p_{25}\}$. This unfairly penalizes $a_2$, resulting in $\zeta_{\text{worst}}(S) = \max\{1/26, 1\} = 1$ and $\zeta_{\text{avg}}(S) = \frac{1}{2}(1/26 + 1) = 27/52$, which is much worse compared with the ideal results. In contrast, our method in Algorithm 1 solves the linear program and recovers the ideal solution, achieving the same fairness metrics as the optimal case.

A simple workaround to mitigate unfairness in conventional desk-rejection systems is the roulette algorithm, which randomly rejects papers from non-compliant authors like $a_1$ until the submission limit $x$ is reached. However, this heuristic cannot fully prevent the rejection of the undesirable paper $p_{26}$ and results in suboptimal fairness outcomes compared to our fairness-aware rejection, since the expected fairness metrics under the roulette algorithm satisfy $\mathbb{E}[\zeta_{\text{worst}}] = (25/26) \cdot (1/26) + (1/26) \cdot 1 \leq 1/26$ and $\mathbb{E}[\zeta_{\text{avg}}] = (25/26) \cdot (1/52) + (1/26) \cdot (27/52) \leq 27/52$.

Thus, this example illustrates that conventional desk-rejection systems in top conferences such as CVPR can suffer from severe fairness issues, whereas our proposed

---

**Algorithm 2** Conventional Desk-Reject Algorithm

1: **procedure** DESKREJCT($\mathcal{A}, \mathcal{P}, x$)
2:     /* Initialize registered paper set for each author. */
3:     **for** $i = 1 \to n$ **do**
4:         $R_i \leftarrow \emptyset$
5:     **end for**
6:     /* Initialize the subset of remaining papers. */
7:     $S \leftarrow \mathcal{P}$
8:     /* Process each paper in submission order.*/
9:     **for** $j = 1 \to m$ **do**
10:         **for** $i \in A_j$ **do**
11:             /* If author $a_i$ has reached the submission limit, the paper will be rejected.*/
12:             **if** $|R_i| \geq x$ **then**
13:                 $S \leftarrow S \setminus \{p_j\}$
14:                 **break**
15:             **end if**
16:         **end for**
17:         /* If paper $p_j$ is not rejected, we add it to each co-author's registered paper set.*/
18:         **if** $p_j \in S$ **then**
19:             **for** $i \in A_j$ **do**
20:                 $R_i \leftarrow R_i \cup \{j\}$
21:             **end for**
22:         **end if**
23:     **end for**
24:     **return** $S$
25: **end procedure**

---

method effectively mitigates these problems.

Additionally, this example also highlights another noteworthy consequence of the conventional desk-rejection system. Specifically, authors collaborating with senior researchers who have numerous submissions may have to compete for earlier submission slots to avoid desk rejection. However, the submission order should not influence whether a paper is accepted, which reveals the unintended implications of the order-based desk-rejection system.

## 7. Experiments

In this section, we conduct an empirical study comparing the conventional desk-rejection algorithm and our novel fairness-aware desk-rejection algorithm.

*Table 2.* Dataset statistics.

| Dataset | # Authors | # Papers |
|---------|-----------|----------|
| ICLR'21 | 7964 | 2954 |
| ICLR'22 | 8507 | 2617 |
| ICLR'23 | 12451 | 3793 |

*Table 3.* The average fairness $\zeta_{\mathrm{avg}}$ of the conventional desk rejection method and our fairness-aware desk rejection method. A lower $\zeta_{\mathrm{avg}}$ indicates better fairness.

| Dataset | Method | $x = 4$ | $x = 6$ | $x = 8$ | $x = 10$ | $x = 12$ | $x = 14$ |
|---|---|---|---|---|---|---|---|
| ICLR'21 | Conventional | 0.112 | 0.059 | 0.033 | 0.021 | 0.013 | 0.009 |
| ICLR'21 | Fair (Ours) | **0.074** | **0.035** | **0.018** | **0.011** | **0.006** | **0.004** |
| ICLR'22 | Conventional | 0.112 | 0.059 | 0.035 | 0.023 | 0.013 | 0.007 |
| ICLR'22 | Fair (Ours) | **0.073** | **0.036** | **0.019** | **0.010** | **0.005** | **0.002** |
| ICLR'23 | Conventional | 0.115 | 0.056 | 0.031 | 0.022 | 0.015 | 0.009 |
| ICLR'23 | Fair (Ours) | **0.074** | **0.033** | **0.018** | **0.011** | **0.007** | **0.004** |

**Datasets.** Most desk-rejection data from real-world conferences (e.g., CVPR, KDD, etc.) is not public and is only available to the conferences' chairs. Thus, we conduct simulation experiments with public ICLR data from the OpenReview API [1]. Specifically, we crawl the data using the invitation link "`ICLR.cc/{year}/Conference/-/Blind_Submission`," selecting three years of data: 2021, 2022, and 2023. This results in our three datasets for evaluation: **ICLR'21**, **ICLR'22**, **ICLR'23**. The statistics of these datasets can be found in Table 2.

Note that some papers may be overlooked by the OpenReview API, so the number of papers and authors included in our evaluation may not fully match the total number of real conference papers. However, we believe this deviation in the number of authors and papers in our crawled dataset does not significantly differ from publicly available statistics of the ICLR conference.

**Baselines.** To the best of our knowledge, we are the first paper to study the fairness of the submission-limit-based desk-rejection problem. Thus, our only baseline is the current desk-rejection algorithm used in major conferences (Algorithm 2), which rejects all papers with non-compliant authors based on submission order. We compare this with our proposed group fairness optimization method (Algorithm 1), which first rejects papers from senior researchers with numerous submissions while protecting junior researchers with fewer submissions.

**Experimental Settings.** For the linear program solver LPSolver in Algorithm 1, we use the standard linear program solver in the Python PuLP library [2]. Before running the solver, to accelerate computation, we first remove all safe authors whose papers do not include any non-compliant authors exceeding the submission limit, as these authors have no risk of desk-rejection. Our source code is available at https://github.com/magiclinux/desk_reject_fairness_icml_2025/.

[1]https://docs.openreview.net/reference/api-v2
[2]https://pypi.org/project/PuLP

**Results.** We set the submission limit to $x \in \{4, 6, 8, 10, 12, 14\}$ and run the desk-rejection algorithm. We use average fairness $\zeta_{\mathrm{avg}}$, as defined in Definition 5.4, as our evaluation metric. Our experimental results are shown in Table 7. We find that, in most cases, our proposed method achieves a significant cost reduction compared to previous methods, with a reduction of approximately 1/3 to 1/2. Thus, we conclude that our proposed fairness-aware desk-rejection method significantly outperforms current desk-rejection algorithms in most major conferences, demonstrating a promising future for improving fairness in AI conferences.

## 8. Conclusion

In this work, we identify the fairness issue in the desk-rejection mechanisms of AI conferences under submission limits. Our theoretical analysis shows that an ideal system that rejects papers solely based on authors' non-compliance, without unfairly penalizing others due to collective punishment, is impossible. We further consider an optimization-based fairness-aware desk-rejection system to alleviate the unfairness problem. In this system, we considered two fairness metrics: worst-case fairness and average fairness, and formally established that optimizing worst-case fairness in desk-rejection is NP-hard, while optimizing average fairness can be reduced to a linear programming problem that can be solved highly efficiently. Through case studies, we showed that the proposed method outperforms the existing systems in top AI conferences.

For future work, it would be interesting to study how strategic behavior reshapes fairness once the system reaches equilibrium. For example, we can cast the submission process as an $n$-player game in which authors select collaborators and project counts, balancing project costs against the rewards of clearing desk rejection and securing acceptance, and then solve for the Nash equilibrium. Moreover, extending this baseline to include real-world factors, such as prior collaborations, idea ownership, and unequal resources, will clarify which alternative desk-rejection rules can sustain equity under fully strategic play.

## Acknowledgements

The author would like to thank the anonymous reviewer of ICML 2025 for their highly insightful suggestions. The work is partially supported by the National Science Foundation (NSF) Grant CCF-2046710.

## Impact Statement

This paper seeks to advance fairness in desk-rejection systems employed by top AI conferences. By formally defining the paper submission limit problem, we demonstrate that an ideal system that rejects papers solely based on excessive submissions without negatively impacting innocent authors is mathematically impossible when multiple authors are involved. To address this, we propose a framework that optimizes desk-rejection fairness based on two different metrics, ensuring that early-career researchers with fewer submissions are less likely to face disproportionate setbacks due to co-authors' submission behavior. While introducing fairness may slightly impact the acceptance rates of senior researchers with a large number of submissions, our approach does not aim to diminish anyone's research output. Instead, it seeks to balance opportunity across career stages, thereby advancing social justice and contributing to a fairer and more inclusive ML research ecosystem.

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

# Appendix

**Roadmap.** In Section A, we supplement the missing proofs in Section 4. In Section B, we present the missing proofs in Section 5. In Section C, we show additional case studies. In Section D, we provide the details related to conference submission limits.

## A. Missing Proofs in Section 4

In this section, we provide the complete technical proofs for Theorem 4.3 in Section 4. In Section A.1, we first introduce key definitions that will be useful To structure our analysis, we in the subsequent proofs. We then establish positive results for the cases where $n \leq 2$ in Section A.2, followed by negative results for $n \geq 3$ in Section A.3.

### A.1. Basic Definitions

To systematically analyze the desk-rejection problem, we begin by classifying authors based on their submission behavior and their relationship to co-authors. This classification will help us organize and present the proofs in a more structured and readable manner.

**Definition A.1** (Author Categories). *For any author $a_i \in \mathcal{A}$, we define the following categories:*

- ***Non-compliant****: An author $a_i$ is non-compliant if they have submitted more than $x$ papers, i.e., $|P_i| > x$. Such authors exceed the submission limit and are subject to desk-rejection under the policy.*

- ***Vulnerable****: An author $a_i$ is vulnerable if they have submitted no more than $x$ papers ($|P_i| \leq x$) but have at least one non-compliant co-author, i.e., $\exists k \in C_i$ such that $|P_k| > x$. Although these authors comply with the submission limit, they are at risk of being unfairly penalized due to their co-authors' non-compliance.*

- ***Safe****: An author $a_i$ is safe if they have submitted no more than $x$ papers ($|P_i| \leq x$) and all their co-authors are also compliant, i.e., $\forall k \in C_i, |P_k| \leq x$. These authors are guaranteed to retain all their submissions, as neither they nor their co-authors violate the submission limit.*

Next, we formalize the notion of achievability for the ideal desk-rejection system.

**Definition A.2** (Achievability). *Given a submission limit problem instance as defined in Definition 3.1:*

- ***Positive result****: A problem instance is a positive result if there exists an algorithm that can achieve the ideal desk-rejection as defined in Definition 4.1.*

- ***Negative result****: A problem instance is a negative result if, under proper conditions, no algorithm can achieve the ideal desk-rejection as defined in Definition 4.1.*

In the following sections, we will use these definitions to systematically prove the positive results for small numbers of authors ($n \leq 2$) and the negative results for larger numbers of authors ($n \geq 3$), which covers two cases in Theorem 4.3.

### A.2. Positive Results

In this subsection, we present two positive results that support the $n \leq 2$ case in Theorem 4.3. We begin with the positive result for $n = 1$ and any $x \in \mathbb{N}_+$.

**Lemma A.3** (Positive Result for $n = 1$ and Any $x \in \mathbb{N}_+$, General Case). *If the following conditions hold:*

- *Let $n = 1$ denote the number of authors as defined in Definition 3.1.*

- *Let $x \in \mathbb{N}_+$ denote the maximum number of submissions allowed for each author in the conference.*

*Then, there exists an algorithm that achieves the ideal desk-rejection as defined in Definition 4.1.*

*Proof.* We consider the three cases for the only author $a_1$: non-compliant, vulnerable, and safe, as defined in Definition A.1.

**Case 1: Non-compliant author.** If author $a_1$ is non-compliant, we desk-reject $(|P_1| - x)$ papers. This ensures that exactly $x$ papers remain, satisfying the ideal desk-rejection condition.

**Case 2: Vulnerable author.** Since $n = 1$ and there is only one author, author $a_1$ has no co-authors to make itself vulnerable. Therefore, this case cannot happen.

**Case 3: Safe author.** If author $a_1$ is safe, no papers need to be rejected. The ideal desk-rejection condition is trivially satisfied.

In all possible cases, we can achieve the ideal desk-rejection. Thus, the proof is finished. $\square$

To present the positive result for $n = 2$ and any $x \in \mathbb{N}_+$, we first discuss a specific case where all authors are non-compliant.

**Lemma A.4** (Positive Result for $n = 2$ and Any $x \in \mathbb{N}_+$, Non-compliant Author Only Case). *If the following conditions hold:*

- *Let $n = 1$ denote the number of authors as defined in Definition 3.1.*

- *All the authors are non-compliant authors as defined in Definition A.1.*

- *Let $x \in \mathbb{N}_+$ denote the maximum number of submissions allowed for each author in the conference.*

*Then, there exists an algorithm that achieves the ideal desk-rejection as defined in Definition 4.1.*

*Proof.* Let $c \in \mathbb{N}$ denote the number of papers co-authored by both author $a_1$ and author $a_2$. For $i \in \{1, 2\}$, let $b_i \in \mathbb{N}$ denote the number of single-authored papers by author $a_i$.

We then have:

$$b_1 + c = |P_1|$$

and

$$b_2 + c = |P_2|.$$

**Case 1: $c \leq x$.** In this case, we have $b_1 \geq |P_1| - x$ and $b_2 \geq |P_2| - x$. Since $b_i$ represents the number of single-authored papers by author $a_i$, we can desk reject exactly $(|P_i| - x)$ papers from author $a_i$.

**Case 2: $c > x$.** Here, we have $b_1 < |P_1| - x$ and $b_2 < |P_2| - x$. We first desk reject all $b_1$ single-authored papers from author $a_1$ and all $b_2$ single-authored papers from author $a_2$. Next, we desk reject $(c - x)$ co-authored papers from both authors. This ensures that the remaining $x$ papers are co-authored by both $a_1$ and $a_2$. Thus, we have successfully rejected exactly $(|P_i| - x)$ papers from each author $a_i$.

By combining the two cases above, the proof is complete. $\square$

With the help of Lemma A.4, we now establish the positive result for $n = 2$ and any $x \in \mathbb{N}_+$.

**Lemma A.5** (Positive Result for $n = 2$ and Any $x \in \mathbb{N}_+$, General Case). *If the following conditions hold:*

- *Let $n = 1$ denote the number of authors as defined in Definition 3.1.*

- *Let $x \in \mathbb{N}_+$ denote the maximum number of submissions allowed for each author in the conference.*

*Then, there exists an algorithm that achieves the ideal desk-rejection as defined in Definition 4.1.*

*Proof.* We consider two authors, $a_1$ and $a_2$. Without loss of generality, we assume that $a_1$ has at least as many papers as $a_2$, i.e., $|P_1| \geq |P_2|$. By exhaustively enumerating all possible compositions of author types (i.e., non-compliant, vulnerable, or safe) for $a_1$ and $a_2$, we observe that the vulnerable-safe composition is impossible. This is because a vulnerable author must co-author at least one paper with a non-compliant author. After excluding this case, we analyze the remaining possible scenarios as follows:

**Case 1: Both $a_1$ and $a_2$ are safe authors.** In this case, no papers need to be rejected, and the ideal desk-rejection trivially holds.

**Case 2: $a_1$ is a non-compliant author and $a_2$ is a safe author.** Since rejecting papers from $a_1$ does not affect $a_2$'s submissions, we can simply reject $(|P_1| - x)$ papers from $a_1$ to achieve the ideal desk-rejection.

**Case 3: $a_1$ is a non-compliant author and $a_2$ is a vulnerable author.** By Definition A.1, we have $|P_1| > x$ and $|P_2| \leq x$. Let $c := |\{p_j \in S : p_j \in P_1, p_j \in P_2\}|$ denote the number of co-authored papers by $a_1$ and $a_2$. From basic set theory, we know that $c \leq |P_2|$. Since $|P_2| \leq x$, it follows that $c \leq x$. Therefore, we have:

$$\underbrace{|P_1| - c}_{\text{Individual papers of } a_1} \quad \geq \quad \underbrace{|P_1| - x}_{\text{Excess papers of } a_1} \quad ,$$

which implies that the number of individual papers authored solely by $a_1$ exceeds the number of over-limit papers for $a_1$. Thus, we can first reject $a_1$'s individual papers without affecting $a_2$'s submissions, thereby achieving the desired ideal desk-rejection.

**Case 4: Both $a_1$ and $a_2$ are non-compliant authors.** This case directly follows from Lemma A.4.

Combining all the cases above, we conclude that the ideal desk-rejection can always be achieved, which finishes the proof. $\square$

### A.3. Negative Results

In this subsection, we present two positive results that support the $n \geq 3$ case in Theorem 4.3. We commence by showing the negative result for $n = 3$ and $x = 1$.

**Lemma A.6** (Negative Result for $n = 3$ and $x = 1$). *If the following conditions hold:*

- *Let $n = 3$ denote the number of authors as defined in Definition 3.1.*

- *Let $x = 1$ denote the maximum number of submissions allowed for each author in the conference.*

*Then, under proper conditions, no algorithm can achieve the ideal desk-rejection as defined in Definition 4.1.*

*Proof.* Let all the authors be non-compliant authors as defined in Definition A.1, and let the number of papers be $m = 3$. We suppose the three papers $p_1$, $p_2$, and $p_3$ have the following authorship:

- Paper $p_1$ is co-authored by $a_1$ and $a_2$.

- Paper $p_2$ is co-authored by $a_1$ and $a_3$.

- Paper $p_3$ is co-authored by $a_2$ and $a_3$.

From the authors' perspective, the relationships are as follows:

- Author $a_1$ has papers $p_1$ and $p_2$.

- Author $a_2$ has papers $p_1$ and $p_3$.

- Author $a_3$ has papers $p_2$ and $p_3$.

We enumerate all possible rejection plans and their outcomes in Table 4.

First, suppose we desk reject paper $p_3$. Then, authors $a_2$ and $a_3$ each have one paper remaining, but author $a_1$ still has two papers. To satisfy the constraint $x = 1$, we must reject one of $p_1$ or $p_2$.

If we reject $p_1$, author $a_2$ is left with no papers, which is unfair. If we reject $p_2$, author $a_3$ is left with no papers, which is also unfair.

Thus, no rejection plan satisfies the ideal desk rejection condition for all authors. This completes the proof. $\square$

*Table 4.* Remaining number of papers for each author after desk rejection.

| Rejected Papers | Author $a_1$ | Author $a_2$ | Author $a_3$ |
|:---:|:---:|:---:|:---:|
| N/A | 2 | 2 | 2 |
| $p_1$ | 1 | 1 | 2 |
| $p_2$ | 1 | 2 | 1 |
| $p_3$ | 2 | 1 | 1 |
| $p_1, p_2$ | 0 | 1 | 1 |
| $p_1, p_3$ | 1 | 0 | 1 |
| $p_2, p_3$ | 1 | 1 | 0 |
| $p_1, p_2, p_3$ | 0 | 0 | 0 |

Next, we present the negative result for any $n \geq 3$ and $x = n - 2$.

**Lemma A.7** (Negative Result for Any $n \geq 3$ and $x = n - 2$). *If the following conditions hold:*

- *Let $n \geq 3$ denote the number of authors as defined in Definition 3.1.*

- *Let $x = n - 2$ denote the maximum number of submissions allowed for each author in the conference.*

*Then, under proper conditions, no algorithm can achieve the ideal desk-rejection as defined in Definition 4.1.*

*Proof.* In this negative problem instance, we choose the number of papers to be the same as the number of authors, i.e., $m = n$, and we assume all the $n$ authors are non-compliant authors as defined in Definition A.1.

For each of the $n$ papers $p_i \in \mathcal{P}$, we let $i$-th paper $p_i$ contain $n - 1$ authors, excluding only the $i$-th author $a_i$. Specifically, we have:

- The first paper $p_1$ has authors $a_2, a_3, \cdots, a_n$.

- The second paper $p_2$ has authors $a_1, a_3, a_4, \cdots, a_n$.

- $\cdots \cdots$

- The $(n-1)$-th paper has authors $a_1, a_2, \cdots, a_{n-2}, a_n$.

- The $n$-th paper has authors $a_1, a_2, \cdots, a_{n-2}, a_{n-1}$.

Since each author is allowed to submit at most $x = n - 2$ papers, we must desk-reject at least two papers. We analyze the process of desk-rejecting these two papers step by step.

**Step 1: Desk-reject the first paper.**

Without loss of generality, we consider rejecting paper $p_1$ first. After this operation, authors $a_2, a_3, \cdots a_n$, will have $n - 2$ submitted papers, while author $a_1$ will have $n - 1$ submitted papers.

**Step 2: Desk-reject the second paper.**

Without loss of generality, we consider rejecting paper $p_2$ next. After this operation, authors $a_3, a_4, \cdots a_n$, will have $n - 3$ submitted papers, while author $a_1$ and $a_2$ will have $n - 2$ submitted papers.

At this point, it is impossible for authors $a_3, a_4, a_5 \cdots, a_n$ to have exactly $(n-2)$ submitted papers. Therefore, no algorithm can achieve the ideal desk-rejection under the given conditions. This completes the proof.

$\square$

# B. Missing Proofs in Section 5

In this section, we first present the missing proofs for fairness metrics in Section B.1, and then present the supplementary proofs for the hardness of worst-case fairness optimization in Section B.2. Finally, we show the additional proofs for the average fairness optimization problem in Section B.3.

## B.1. Fairness Metrics

We present the relationship between the fairness metrics.

**Proposition B.1** (Relationship of Fairness Metrics, formal version of Proposition 5.6 in Section 5.1)**.** *For any solution* $S \subseteq \mathcal{P}$ *for the submission limit problem in Definition 3.1, we have*

$$\zeta_{\mathrm{avg}}(S) \leq \zeta_{\mathrm{worst}}(S).$$

*Proof.* By Definition 5.5, we have:

$$\zeta_{\mathrm{avg}}(S) = \frac{1}{n} \sum_{i \in [n]} c(a_i, S)$$

$$\leq \frac{1}{n} \sum_{i \in [n]} \max_{i \in [n]} c(a_i, S)$$

$$= \frac{1}{n} \cdot n \cdot \max_{i \in [n]} c(a_i, S)$$

$$= \zeta_{\mathrm{worst}}(S),$$

where the first equality directly follows from Definition 5.5, the second and the third inequality follow from basic algebra, and the last equality follows from Definition 5.4. Thus, we complete the proof. $\square$

## B.2. Hardness of Worst-Case Fairness-Aware Submission Limit Problem

Before proving the theoretical results in Section 5.2, we first introduce a useful fact that serves as a foundation for the subsequent proofs.

**Fact B.2.** *For each author* $a_i \in \mathcal{A}$*, the number of papers after desk-rejection (i.e.,* $|\{p_j \in S : a_i \in A_j\}|$*) can be written as* $W_i^\top r$*.*

*Proof.* This simply follows from:

$$W_i^\top r = \sum_{j \in [m]} W_{i,j} \cdot r_j$$

$$= |\{j \in [m] : W_{i,j} = 1, r_j = 1\}|$$

$$= |\{p_j \in \mathcal{P} : a_i \in A_j, p_j \in S\}|$$

$$= |\{p_j \in S : a_i \in A_j\}|,$$

where the first and the second equality follow from basic algebra and set theory, and the third and the fourth equality follow from Definition 3.1. $\square$

With the help of the aforementioned fact, we now prove the equivalence of the matrix form for the worst-case fairness problem.

**Proposition B.3** (Matrix Form Equivalence for $\zeta_{\mathrm{worst}}$, formal version of Proposition 5.10 in Section 5.2)**.** *The worst-case fairness-aware submission limit problem in Definition 5.7 and the matrix form integer programming problem in Definition 5.9 are equivalent.*

*Proof.* In Definition 5.7, the paper set $\mathcal{P}$ consists of $m$ papers, each of which can either be maintained or desk-rejected. Thus, the subset of maintained papers, $\mathcal{S}$, can be represented by a 0-1 vector $r \in \{0, 1\}^m$, where $r_j = 1$ indicates that paper $p_j$ is maintained, and $r_j = 0$ indicates that it is desk-rejected. We now establish the equivalence of both the objective function and the constraints in these two formulations.

**Part 1: Optimization Objective.** We first consider the objective function $\mathbf{1}_n^\top D^{-1} W r$ in Definition 5.9:

$$
\begin{aligned}
\min_{r \in \{0,1\}^m} \|\mathbf{1}_n - D^{-1} W r\|_\infty &= \min_{r \in \{0,1\}^m} \max_{i \in [n]} (1 - (D^{-1} W r)_i) \\
&= \min_{r \in \{0,1\}^m} \max_{i \in [n]} (1 - (W_i^\top r)_i / D_{i,i}) \\
&= \min_{r \in \{0,1\}^m} \max_{i \in [n]} (1 - (W_i^\top r)_i / |P_i|) \\
&= \min_{r \in \{0,1\}^m} \max_{i \in [n]} (1 - |\{p_j \in S : a_i \in A_j\}| / |P_i|) \\
&= \min_{r \in \{0,1\}^m} \max_{i \in [n]} c(a_i, S) \\
&= \min_{r \in \{0,1\}^m} \zeta_{\text{worst}}(S),
\end{aligned}
$$

where the first equality follows from the definition of infinity norm, the second equality follows from basic algebra, the third equality follows from Definition 5.9, the fourth equality follows from Fact B.2, the fifth equality follows from Definition 5.1, and the last equality follows from Definition 5.4. By decoding $r$ back into the paper subset $S$, we recover the original optimization objective in Definition 5.7.

**Part 2: Constraints.** The constraint in Definition 5.9 can be rewritten using basic algebra as:

$$
W_i \cdot r \leq x, \quad \forall i \in [n].
$$

By applying Fact B.2, we see that this constraint is equivalent to its counterpart in Definition 5.7.

Since both the objective function and constraints in Definition 5.7 and Definition 5.9 are equivalent, the proof is complete. $\square$

To show the hardness of the worst-case fairness problem, we first present a classical set cover problem with well-established hardness.

**Definition B.4** (Set Cover Problem (Karp, 1972; Garey & Johnson, 1979)). *The Set Cover problem is the following:*

- **Input:** *A universe $U = \{1, \ldots, n\}$, a family of sets $\{S_1, \ldots, S_m\} \subseteq 2^U$, and a integer $K > 0$.*

- **Question:** *Is there a subfamily $\{S_j : j \in J\}$ for some $J \subseteq \{1, \ldots, m\}$ and $|J| \leq K$ that covers $U$, i.e., $\bigcup_{j \in J} S_j = U$?*

**Lemma B.5** (Folklore (Karp, 1972; Garey & Johnson, 1979)). *The Set Cover problem defined in Definition B.4 is* NP-hard.

Additionally, we also present a technical lemma which is useful for showing the hardness of the worst-case fairness problem.

**Lemma B.6.** *For any $r \in \{0,1\}^m$, the following two statements are equivalent:*

- **Part 1.** $\|\mathbf{1}_n - D^{-1} W r\|_\infty \leq 1 - \frac{1}{\min_{i \in [n]} |P_i|}$.

- **Part 2.** $\min_{i \in [n]} (W r)_i \geq 1$.

*Proof.* We first show that Part 1 implies Part 2. Suppose that

$$
\|\mathbf{1}_n - D^{-1} W r\|_\infty \leq 1 - \frac{1}{\min_{i \in [n]} |P_i|}.
$$

By the definition of the infinity norm, we have

$$
1 - \frac{(W r)_{i'}}{|P_{i'}|} \leq 1 - \frac{1}{\min_{i \in [n]} |P_i|}, \quad \forall i' \in [n].
$$

Rearranging gives

$$
(W r)_{i'} \geq \frac{|P_{i'}|}{\min_{i \in [n]} |P_i|} \geq 1, \quad \forall i' \in [n].
$$

Since for all $i' \in [n]$, we have $(Wr)_{i'} \geq 1$, we can conclude that $\min_{i \in [n]} (Wr)_i \geq 1$.

Now we show that that Part 2 implies Part 1. Suppose that $\min_{i \in [n]} (Wr)_i \geq 1$, then we have $(Wr)_i \geq 1$ for all $i \in [n]$, which implies that for all $i \in [n]$,

$$1 - \frac{(Wr)_i}{|P_i|} \leq 1 - \frac{1}{|P_i|} \leq 1 - \frac{1}{\max_{i' \in [n]} |P_{i'}|}.$$

Hence

$$\|\mathbf{1}_n - D^{-1} Wr\|_\infty \leq 1 - \frac{1}{\min_{i \in [n]} |P_i|}.$$

Thus the proof is complete. $\qquad\square$

**Theorem B.7** (Hardness, formal version of Theorem 5.11 in Section 5.2). *The Worst-Case Fairness-Aware Submission Limit Problem defined in Definition 5.7 is* NP-hard.

*Proof.* By Proposition 5.10, it suffices to reduce the Set Cover problem to the integer optimization problem of the matrix form in Definition 5.9.

Given an instance of Set Cover, we build the matrix $W \in \{0,1\}^{n \times m}$ by defining $W_{i,j} = 1$ if element $i \in S_j$, and $0$ otherwise. Now set $|P_i| = \sum_{j \in [m]} W_{i,j}$ for every row $i \in [n]$. Finally, we choose $x = m$. We reduce the Set Cover problem to the following optimization problem:

$$\min_{r \in \{0,1\}^m} \|\mathbf{1}_n - D^{-1} Wr\|_\infty$$
$$\text{s.t. } Wr \leq m\mathbf{1}_n,$$
$$\|r\|_1 \leq K.$$

Note that this problem is easier than the optimization problem defined in Definition 5.7. The constraint $Wr \leq m\mathbf{1}_n$ is always satisfied, so we can drop it out. Now, it suffices to consider the decision problem:

$$\text{Find } r \in \{0,1\}^m$$
$$\text{s.t. } \|\mathbf{1}_n - D^{-1} Wr\|_\infty \leq 1 - \frac{1}{\min_{i \in [n]} |P_i|},$$
$$\|r\|_1 \leq K.$$

Note that $\|\mathbf{1}_n - D^{-1} Wr\|_\infty \leq 1 - \frac{1}{\min_{i \in [n]} |P_i|}$ is equivalent to $\min_{i \in [n]} (Wr)_i \geq 1$ by Lemma B.6.

Hence the problem is equivalent to

$$\text{Find } r \in \{0,1\}^m \quad \text{s.t.} \quad \min_{i \in [n]} (Wr)_i \geq 1 \text{ and } \|r\|_1 \leq K.$$

It is not hard to see that the Set Cover problem has a solution if and only if the above problem has a solution. Requiring $\min_{i \in [n]} (Wr)_i > 1$ exactly means that each element $i$ in the universe is covered by at least set $S_j$. The constraint $\|r\|_1 \leq K$ means that the size of cover is at most $K$. In other words, there exists a subfamily of size at most $K$ covering all elements if and only if there is an $r \in \{0,1\}^m$ with $\min_{i \in [n]} (Wr)_i > 1$ and $\|r\|_1 \leq K$.

Therefore, by Lemma B.5, the worst-case fairness-aware submission limit problem is NP-hard. $\qquad\square$

### B.3. Average Fairness Optimization

Now, we present the missing proofs on both matrix form equivalence and linear programming optimal solution equivalence for the average fairness optimization problem.

**Proposition B.8** (Matrix Form Equivalence for $\zeta_{\text{avg}}$, formal version of Proposition 5.14 in Section 5.3). *The problem in Definition 5.12 and the problem in Definition 5.13 are equivalent.*

*Proof.* In Definition 5.12, there are $m$ papers in $\mathcal{P}$, where each paper can either be maintained or rejected. Thus, we can encode the paper subset $S$ using a binary vector $r \in \{0,1\}^m$, where $r_j = 1$ indicates that paper $p_j$ is maintained, and $r_j = 0$ indicates that it is desk-rejected. We now demonstrate that both the objective function and the constraints are equivalent.

**Part 1: Optimization Objective.** We first examine the objective function $\mathbf{1}_n^\top D^{-1} W r$ in Definition 5.13:

$$
\begin{aligned}
\mathbf{1}_n^\top D^{-1} W r &= \sum_{i \in [n]} (D^{-1} W r)_i \\
&= \sum_{i \in [n]} (W \cdot r)_i / |P_i| \\
&= \sum_{i \in [n]} (W_i^\top \cdot r) / |P_i| \\
&= \sum_{i \in [n]} \frac{|\{p_j \in S : a_i \in A_j\}|}{|P_i|} \\
&= \sum_{i \in [n]} (1 - c(a_i, S)),
\end{aligned}
$$

where the first equality follows from basic algebra, the second follows from Definition 5.13, the third follows from matrix-vector multiplication, the fourth follows from Fact B.2, and the final equality follows from Definition 5.1. Consequently, the maximization problem in Definition 5.13 can be rewritten as:

$$
\max_{r \in \{0,1\}^m} \sum_{i \in [n]} (1 - c(a_i, S)).
$$

Since maximizing this objective is equivalent to minimizing $\sum_{i \in [n]} c(a_i, S)$, we can reformulate it as:

$$
\min_{r \in \{0,1\}^m} \sum_{i \in [n]} c(a_i, S).
$$

By decoding $r$ back into the paper subset $S$, we recover the original optimization objective in Definition 5.12.

**Part 2: Constraints.** Since the constraint is identical to that in the worst-case fairness minimization problem in Definition 5.7, this result follows directly from Part 2 in the proof of Proposition B.3.

Since both the objective function and constraints in Definition 5.12 and Definition 5.13 are equivalent, the proof is complete. □

**Theorem B.9** (Optimal Solution Equivalence of the Relaxed Problem, formal version of Theorem 5.16 in Section 5.3). *The optimal solution of the relaxed problem in Definition 5.15 is equivalent to the optimal solution of the original problem in Definition 5.13.*

*Proof.* The problem in Definition 5.15 is a linear program since it has a linear objective function $\mathbf{1}_n^\top D^{-1} W r$ and linear constraints: the box constraint $r \in [0, 1]^m$ and a linear inequality constraint $(Wr)/x \le \mathbf{1}_n$.

Furthermore, the problem is convex because the objective function is linear, the constraint $(Wr)/x \le \mathbf{1}_n$ is affine, and the feasible region defined by $r \in [0, 1]^m$ is a convex set.

By the fundamental theorem of linear programming (see Page 23 of (Luenberger & Ye, 1984)), the optimal solution must occur at an extreme point of the convex polytope defined by the constraints. This implies that for all $i \in [m]$, we must have either $r_i = 0$ or $r_i = 1$. Therefore, the optimal solution of the relaxed linear program coincides with that of the original integer program, which finishes the proof. □

## C. Additional Case Studies

As discussed in Section 4.2, optimizing the worst-case fairness metric is computationally challenging. Therefore, we minimize the average fairness metric, which serves as a lower bound for worst-case fairness, as a practical alternative. In this subsection, we present case studies demonstrating the relationship between both types of fairness metrics.

**Example C.1.** *Consider a submission limit problem as defined in Definition 3.1 with $x = 2$, $n = 3$ authors, and $m = 6$ papers. Let author $a_1$ submit four papers $p_1, p_2, p_3, p_4$, author $a_2$ submit two papers $p_3, p_5$, and author $a_3$ submit two papers $p_4, p_6$.*

In this case, the ideal desk-rejection criteria in Definition 4.1 reject $p_1$ and $p_2$ (i.e., $S = \{p_3, p_4, p_5, p_6\}$), yielding fairness metrics $\zeta_{\text{worst}}(S) = \max\{1/2, 0, 0\} = 1/2$ and $\zeta_{\text{avg}}(S) = \frac{1}{3}(1/2 + 0 + 0) = 1/6$. By applying an LP solver to minimize average fairness using Algorithm 1 and enumerating all rejection strategies to verify worst-case fairness minimization, we observe that minimizing average fairness in this case aligns with minimizing worst-case fairness as defined in Definition 5.9. This case illustrates that minimizing average fairness can sometimes benefit worst-case fairness.

However, average fairness and worst-case fairness are not always consistent. In some cases, prioritizing average fairness may disproportionately burden certain individuals. To illustrate this, we consider the following example.

**Example C.2.** *Consider a submission limit problem as defined in Definition 3.1 with $x = 2$, $n = 5$ authors, and $m = 4$ papers. Let author $a_1$ submit four papers $p_1, p_2, p_3, p_4$, author $a_2$ submit two papers $p_1, p_2$, and authors $a_3, a_4, a_5$ be coauthors of papers $p_3, p_4$.*

In this scenario, an ideal desk-rejection is impossible because $a_1$ must have two papers rejected, but rejecting any papers would cause at least one of the authors in $a_2, \ldots, a_5$ to fall below the submission limit of $x = 2$. Here, average fairness and worst-case fairness diverge: Algorithm 1 minimizes average fairness by rejecting $p_1$ and $p_2$ (i.e., $S = \{p_3, p_4\}$), which unfairly excludes all of $a_2$'s submissions. This results in fairness metrics $\zeta_{\text{avg}}(S) = \frac{1}{4}(1/2 + 1 + 0 + 0) = 3/8$ and $\zeta_{\text{worst}}(S) = \max\{1/2, 1, 0, 0\} = 1$.

Conversely, the worst-case fairness minimization problem in Definition 5.9 rejects one paper from $a_1, a_2$ and another from $a_3, a_4$, leading to $\zeta_{\text{avg}}(S) = \frac{1}{4}(1/2 + 1/2 + 1/2 + 1/2) = 1/2$ and $\zeta_{\text{worst}}(S) = \max\{1/2, 1/2, 1/2, 1/2\} = 1/2$.

This example highlights an unintended consequence of minimizing average fairness: it may unfairly penalize authors with fewer coauthors, as rejecting their papers incurs a smaller total cost. On the other hand, optimizing worst-case fairness inevitably spreads rejections across a broader set of authors, potentially leading to a higher overall fairness cost. Balancing individual and average fairness remains an open challenge, which we leave for future work.

## D. Summary of Conference Links

In the introduction, Table 1 only gives a brief summary of the conference year and its limitation of per-author submission. Thus, we provide a detailed list of conferences in each year in this section, and then summarize the submission limits in Table. 5.

- CVPR
  - 2025, https://cvpr.thecvf.com/Conferences/2025/CVPRChanges
  - 2024, https://cvpr.thecvf.com/Conferences/2024/AuthorGuidelines

- ICCV
  - 2025, https://iccv.thecvf.com/Conferences/2025/AuthorGuidelines
  - 2023, https://iccv2023.thecvf.com/policies-361500-2-20-15.php

- AAAI
  - 2025, https://aaai.org/conference/aaai/aaai-25/submission-instructions/
  - 2024, https://aaai.org/aaai-24-conference/submission-instructions/
  - 2023, https://aaai-23.aaai.org/submission-guidelines/
  - 2022, https://aaai.org/conference/aaai/aaai-22/

- WSDM
  - 2025, https://www.wsdm-conference.org/2025/call-for-papers/
  - 2024, https://www.wsdm-conference.org/2024/call-for-papers/
  - 2023, https://www.wsdm-conference.org/2023/calls/call-papers/

  – 2022, https://www.wsdm-conference.org/2022/calls/
  – 2021, https://www.wsdm-conference.org/2021/call-for-papers.php
  – 2020, https://www.wsdm-conference.org/2020/call-for-papers.php

- IJCAI

  – 2025, https://2025.ijcai.org/call-for-papers-main-track/
  – 2024, https://ijcai24.org/call-for-papers/
  – 2023, https://ijcai-23.org/call-for-papers/
  – 2022, https://ijcai-22.org/calls-papers
  – 2021, https://ijcai-21.org/cfp/index.html
  – 2020, https://ijcai20.org/call-for-papers/index.html
  – 2019, https://www.ijcai19.org/call-for-papers.html
  – 2018, https://www.ijcai-18.org/cfp/index.html
  – 2017, https://ijcai-17.org/MainTrackCFP.html

- KDD

  – 2025, https://kdd2025.kdd.org/research-track-call-for-papers/
  – 2024, https://kdd2024.kdd.org/research-track-call-for-papers/
  – 2023, https://kdd.org/kdd2023/call-for-research-track-papers/index.html

*Table 5.* In this table, we summarize the submission limits of top conferences in recent years. For details of each conference website, we refer the readers to Section D in Appendix.

| Conference Name | Year | Upper Bound |
|:---:|:---:|:---:|
| CVPR | 2025 | 25 |
| CVPR | 2024 | N/A |
| ICCV | 2025 | 25 |
| ICCV | 2023 | N/A |
| AAAI | 2025 | 10 |
| AAAI | 2024 | 10 |
| AAAI | 2023 | 10 |
| AAAI | 2022 | N/A |
| WSDM | 2025 | 10 |
| WSDM | 2024 | 10 |
| WSDM | 2023 | 10 |
| WSDM | 2022 | 10 |
| WSDM | 2021 | 10 |
| WSDM | 2020 | N/A |
| IJCAI | 2025 | 8 |
| IJCAI | 2024 | 8 |
| IJCAI | 2023 | 8 |
| IJCAI | 2022 | 8 |
| IJCAI | 2021 | 8 |
| IJCAI | 2020 | 6 |
| IJCAI | 2019 | 10 |
| IJCAI | 2018 | 10 |
| IJCAI | 2017 | N/A |
| KDD | 2025 | 7 |
| KDD | 2024 | 7 |
| KDD | 2023 | N/A |

