# OpenReview forum: "Dissecting Submission Limit in Desk-Rejections: A Mathematical Analysis of Fairness in AI Conference Policies"
_ICML.cc/2025/Conference — ICML 2025 poster_

### Official Review · Reviewer_XdHw · 2025-03-10

**Overall Recommendation:** 3

**Summary:**

In this paper, the authors highlight that random desk rejection based on per-author submission limits might be unfair. They propose individual and group unfairness definitions to make the AI conference desk rejection policy more fair. The authors propose an LP optimization algorithm to reduce group unfairness (as they show that reducing individual unfairness is computationally intractable). They find that their algorithm outperforms existing policies.

**Claims And Evidence:**

- The paper is well-written and easy to read, though a lot of the theoretical findings/justifications seem to be in the appendix
- the group unfairness definition is confusing: are the definitions applied for each group in each case? or should there be a maximum taken across all groups (similar to the individual unfairness definition)? or is it assumed that all individuals belong to a single group? in that case, it is unclear how group unfairness differs from average fairness
- The utility of the approach is evaluated through a case study. However, the authors could have considered simulation setups, and/or datasets where decisions are public with different simulated policies (e.g. ICLR)

**Essential References Not Discussed:**

The authors do not connect to prior works in fair selection and quota based selection (e.g. https://dl.acm.org/doi/abs/10.1145/3391403.3399482, https://arxiv.org/pdf/2204.03046). A sentence summarizing this literature field and explaining how the author's work differs would be helpful.

**Experimental Designs Or Analyses:**

- The experimental design makes sense, but is limited. The authors take a case-study based evaluation appraoch, but using more simulation studies would have helped test the validity of the algorithm. They only compare to a simple baseline (which is the current desk rejection approach in CVPR)
-  Further, approximate solutions for individual unfairness could have been considered.
- the group unfairness definition is confusing: are the definitions applied for each group in each case? or should there be a maximum taken across all groups (similar to the individual unfairness definition)? or is it assumed that all individuals belong to a single group? in that case, it is unclear how group unfairness differs from average fairness

**Methods And Evaluation Criteria:**

- The utility of the approach is only evaluated through a case study. However, the authors could have considered simulation setups, and/or datasets where decisions are public with different simulated policies (e.g. ICLR). More experimental validation would strengthen the paper.
- The group unfairness definition is confusing when there are multiple groups, and this has not been studied in the experimental validation section

**Other Comments Or Suggestions:**

- Proposition 5.6 (order of group and individual unfairness seems incorrect): have the authors accidentally switched the signs? the proof seems correct in Appendix B

**Other Strengths And Weaknesses:**

*strengths*
- The paper is well-written and easy to read, though a lot of the content seems to be in the appendix
- the approach seems intuitive and is solving an important problem

*weaknesses*
- the group unfairness definition is confusing: are the definitions applied for each group in each case? or should there be a maximum taken across all groups (similar to the individual unfairness definition)? or is it assumed that all individuals belong to a single group? in that case, it is unclear how group unfairness differs from average fairness
- The utility of the approach is evaluated through a case study. However, the authors could have considered simulation setups, and/or datasets where decisions are public with different simulated policies (e.g. ICLR)

**Questions For Authors:**

- the group unfairness definition is confusing: are the definitions applied for each group in each case? or should there be a maximum taken across all groups (similar to the individual unfairness definition)? or is it assumed that all individuals belong to a single group? in that case, it is unclear how group unfairness differs from average fairness
- can authors describe how their work differs from fair selection (e.g. under quotas)?

**Relation To Broader Scientific Literature:**

- The results relate to broader literature in fair selection and fairness in general. It might be worth linking the paper to related works in fair selection (e.g. https://dl.acm.org/doi/abs/10.1145/3391403.3399482 )

**Theoretical Claims:**

- Proposition 5.6 (order of group and individual unfairness seems incorrect): have the authors accidentally switched the signs? the proof seems correct in Appendix B
- It is unclear how the approach works for individuals belonging to multiple groups

---

> ### Author Rebuttal · Authors · 2025-04-01
>
> We sincerely appreciate the reviewers' acknowledgement of this paper’s writing quality and real-world impact. Below, we provide clarifications addressing the weaknesses and questions:
>
> ### **Weakness 1 & Question 1: Group Fairness Definition**
>
> Our definitions of group fairness are inspired by utilitarian social welfare, which measures the average desk-rejection damage across all authors (see lines 282–283 on page 6). This differs from the conventional notion of group fairness, which assesses fairness between distinct groups. In light of this, we plan to rename our “group fairness” metric to “average fairness” (or “entire fairness”) in the next version to avoid confusion.
>
> ### **Weakness 2: Simulation Setups**
>
> Although direct testing on real backend desk-rejection data is difficult, we have added evaluation on our approach using public ICLR (2021–2023) data from the OpenReview API. We compared the current desk-rejection method (Algorithm 2, p.8), with our proposed fairness-aware method (Algorithm 1, p.7). Using the Python PuLP library with submission limits $x=[4,6,8,10,12,14]$, we obtained the results shown below:
>
> **Table R.1: ICLR ’21 Results**
>
> | Method / Limit $x$|4|6| 8 |10|12|14|
> |---------------------------------|-------|-------|-------|-------|-------|-------|
> | Cur. Desk. Rej.| 0.112 | 0.059 | 0.033 | 0.021 | 0.013 | 0.009 |
> | Fair Desk. Rej. | 0.074 | 0.035 | 0.018 | 0.011 | 0.006 | 0.004 |
>
> **Table R.2: ICLR ’22 Results**
>
> |Method / Submission Limit $x$ |4 |6 | 8     | 10    | 12    | 14    |
> |---------------------------------|-------|-------|-------|-------|-------|-------|
> |Cur. Desk. Rej. | 0.112 | 0.059 | 0.035 | 0.023 | 0.013 | 0.007 |
> |Fair Desk. Rej. | 0.073 | 0.036 | 0.019 | 0.010 | 0.005 | 0.002 |
>
> **Table R.3: ICLR ’23 Results**
>
> | Method / Limit $x$ | 4     | 6     | 8     | 10    |12 |14|
> |---------------------------------|-------|-------|-------|-------|-------|-------|
> | Cur. Desk. Rej. | 0.115 | 0.056 | 0.031 | 0.022 | 0.015 |0.009|
> | Fair Desk. Rej. | 0.074 | 0.033 | 0.018 | 0.011 | 0.007 |0.004|
>
> Here, we take the group fairness metric (Definition 5.5), which represents the average cost function (Definition 5.2) across all authors. For instance, a value of 0.02 indicates that on average an author experiences a 2% paper desk-rejection rate. For more details, please refer to W2 of our response to Reviewer UGt2.
>
> ### **Weakness 3: Approximate Solution for Individual Fairness**
>
> In practice, one could relax the discrete constraints to make $r$ continuous and use softmax to substitute the infinity norm, leading to an alternative objective:
> $$\min_{r \in [0,1]^m} \mathsf{Softmax}(1_n - D^{-1}Wr) \quad \text{s.t.} \quad \frac{Wr}{x} \leq 1_n.$$
> This formulation allows for smooth optimization with standard convex solvers. We note that the minimizer of this relaxed objective is not guaranteed to be optimal for the original problem, and we leave a detailed empirical evaluation of this approximation for future work. Additionally, existing mixed programming solvers can handle some medium-scale and non-worst-case problems. For additional discussion, please refer to W3 of our response to Reviewer UGt2.
>
> ### **Weakness 4: Proposition 5.6**
>
> We thank the reviewer for pointing this out. We agree that the proof in Appendix B is correct, and the version in the main text contains a typo where $\leq$ should be $\geq$. We will correct this in our next revision.
>
> ### **Weakness 5: Baselines**
>
> We appreciate the reviewer’s concern regarding baselines. The number of baselines is limited because we are addressing a novel problem that, to the best of our knowledge, has not been studied before (its novelty has also been acknowledged by Reviewer UGt2 and Reviewer q5TE). Our chosen desk-rejection method serves as the only existing baseline and is widely adopted in top conferences, not just CVPR but also other conferences like KDD.
>
> ### **Question 2: Difference Compared with Fair Selection Under Quotas**
>
> Thanks for introducing these relevant works, and we will cite them in future revisions. We believe these works are quite different from ours. First, considering [EGGL’20], they introduce a two-group setting where candidates are selected from two different groups, and the selection rates from these groups are forced to be close. In contrast, in our paper, we use the term group fairness in Definition 5.5 and individual fairness in Definition 5.4 to denote the average or maximum cost function across all authors without any pre-defined group division. Besides, although we also have a quota (i.e., paper submission limit) similar to the ranking fairness paper [YXA’23], we do not incorporate mechanisms such as ranking, consumer probability, or examination probability. This means our setting is fundamentally different.
>
> ### **References**
>
> [EGGL’20] On Fair Selection in the Presence of Implicit Variance. EC 2020.
>
> [YXA’23] Vertical Allocation-based Fair Exposure Amortizing in Ranking. SIGIR 2023.

---

### Official Review · Reviewer_q5TE · 2025-03-10

**Overall Recommendation:** 3

**Summary:**

This paper studies the problem of fairly desk-rejecting papers from conferences, where some of the authors have exceeded per-author submission limits. The paper establishes that this can’t be done without desk-rejecting papers from authors who haven’t violated the limit (since their co-authors might have violated the limit), and develops algorithms for reallocating the costs of desk-rejection away from authors with fewer submissions.

**Claims And Evidence:**

The key claims in this paper:

- Desk-rejecting based on author submission limits must generate unfair collateral damage.
- Finding individually-fail solutions that minimize the maximum cost of desk-rejection to authors is NP hard.
- The authors’ LP-based solution to minimizing the average cost succeeds in improving the fairness of desk-rejections.

are well supported by the arguments presented in the paper.

**Essential References Not Discussed:**

No

**Experimental Designs Or Analyses:**

The authors use a case study, rather than an "experiment", to demonstrate the effect of their proposed system on desk reject. This is sufficient to make their case -- further experiments or analyses would not have provided additional insight.This paper studies the problem of fairly desk-rejecting papers from conferences, where some of the authors have exceeded per-author submission limits. The paper establishes that this can’t be done without desk-rejecting papers from authors who haven’t violated the limit (since their co-authors might have violated the limit), and develops algorithms for reallocating the costs of desk-rejection away from authors with fewer submissions.

**Methods And Evaluation Criteria:**

Yes

**Other Comments Or Suggestions:**

NA

**Other Strengths And Weaknesses:**

The only real problem with this paper is that it doesn't consider the equilibrium effects of desk-rejection policies.

For example, why would authors submit more than the allowed number of papers in the first place?

If having a paper desk-rejected is catastrophic for a junior researcher but merely inconvenient for their senior co-author, then maybe they’ll mutually choose to exclude the senior author from the author list (following norms more common in other disciplines like economics).

On the other hand, if having fewer submissions reduced your chance of having a paper desk rejected, could we see senior researchers working with fewer junior authors? Wouldn’t this also have negative effects on the careers of junior researchers?

It is my strong suspicion that these effects dominate. In other words, the main effect of author submission limit policies is that they change *which* papers are submitted (and with which co-authors). I understand that this problem becomes much harder once one considers the equilibrium, but I don't think a solution is practical unless it does.

**Questions For Authors:**

See above: how could you incorporate equilibrium effects into your approach? I suspect this is too big a question for a rebuttal, though I would raise my score substantially if a compelling answer was given.

**Relation To Broader Scientific Literature:**

To my knowledge the fairness of these desk-rejection methods has not been studied before.

**Theoretical Claims:**

I did not thoroughly check the proofs.

---

> ### Author Rebuttal · Authors · 2025-04-01
>
> We would like to thank the reviewer for acknowledging the novelty of our work and the strong support for our claims. We are pleased to further clarify the motivation behind our paper and to discuss the equilibrium perspective.
>
> ### **Weakness 1 & Question 1: Equilibrium Effects**
>
> Thank you for this interesting and inspiring comment. We believe that the issue raised is highly worthwhile for future study. However, it relates more to the motivation behind the problem than to the flaw in our technical contributions.
>
> 1) Our work demonstrates that even before reaching equilibrium, a submission-limit desk-rejection policy can lead to unfair outcomes. The gaming toward equilibrium mentioned in your comment does not mitigate this unfairness but may, in fact, worsen it. Therefore, we propose alternative approaches that may address the fairness issue.
>
> 2) We agree that in the long term, gaming toward equilibrium might occur after applying a fairness-aware desk-rejection strategy. This is an important question for analyzing the effects of different policies at equilibrium, and we leave a detailed study of this topic for future work. We hope that our current work, as the first analysis of the fairness of desk-rejection methods, serves as a strong starting point.
>
> 3) To incorporate equilibrium effects into this work, we can formally define the problem as follows. Consider that there are $n$ authors. Each author must decide which co-authors to collaborate with and how many projects to pursue. The cost represents the time or resources required for each project. The naive reward is that a paper is not desk rejected, while the final reward is the paper being accepted. The key question is what the optimal strategy is for each author in selecting collaborators and projects. We acknowledge that this formulation omits several realistic considerations, such as relationships among authors, project idea ownership, resource allocation, and more. Solving for the Nash equilibrium of this game would require a more detailed and concrete formulation, which is beyond the scope of the current paper and rebuttal. Nonetheless, we recognize this as an interesting direction for future work.
>
> ### **Weakness 2: Why Authors Submit More Papers Than the Limit**
>
> This phenomenon is common among well-known professors who supervise many students and often do not have the time to verify whether the submission limit is exceeded. They may not be aware of the exact number of papers they have submitted. For instance, in conferences like CVPR and ICCV, a 25-paper limit is sometimes necessary to manage submissions, which highlights how busy these prominent professors can be.
>
> ### **Weakness 3: Excluding Senior Authors from the Author List**
>
> Different research communities have varying traditions regarding authorship. In some cases, it is difficult to exclude senior researchers because they may have provided essential funding or other support. Excluding them could lead to lab politics and harm the relationships between junior and senior researchers, with potentially severe consequences, such as affecting recommendation letters or graduation permissions.
>
> ### **Weakness 4: Senior Researchers Working with Fewer Junior Researchers**
>
> Most projects require significant hands-on contributions, which senior authors often cannot provide due to time constraints. To maximize their outcomes, senior researchers typically prefer to collaborate with junior researchers on each project. Additionally, many senior researchers have a responsibility to work with junior colleagues, such as PhD advisors with their students or tech leads with their engineers.
>
> In this paper, we examine the desk-rejection problem from a conference organizer’s perspective, where over-submission has already occurred. Our goal is to minimize the cost imposed on junior researchers, thereby protecting them from the negative impacts of submission-limit violations.

---

> > ### Comment · Reviewer_q5TE · 2025-04-05
> >
> > I'll stick with my weak accept recommendation.
> >
> > > They may not be aware of the exact number of papers they have submitted.
> >
> > If they're so casually ignorant of the papers they're submitting I might hope they don't mind being left off the author list of a few!
> >
> > > it is difficult to exclude senior researchers
> >
> > This is true. Maybe these limits will help shift the norms in the field.

---

> > > ### Author Response · Authors · 2025-04-05
> > >
> > > Thank you for your positive evaluation and valuable suggestions! Yes, we hope our work may shed light on the AI conference submission system design to create a better and more fair research environment. We appreciate your insightful discussion.

---

### Official Review · Reviewer_UGt2 · 2025-03-11

**Overall Recommendation:** 4

**Summary:**

This paper discusses an interesting fairness issue that occurs in AI conference paper submission scenarios and reveals that the current desk-rejection policy (reject papers when submission limits are exceeded) can unfairly disadvantage early-career researchers, whose submissions may be rejected due to senior co-authors exceeding the paper count limitations. Based on this, the authors first propose the definition of the cost for the author when the paper is rejected, and then provide two fairness definitions based on the cost function, and finally build two optimization frameworks for individual and group fairness, respectively.

## Update after rebuttal
The authors have addressed my concerns. Regarding the supplementary empirical evidence and the novelty of this paper, I have changed my recommendation from 'Weak Accept' to ‘Accept'. I believe this paper will provide valuable insights for future related work.

**Claims And Evidence:**

This paper makes three main claims:

- An ideal system that rejects papers solely based on each author's excessive submissions is mathematically impossible when there are more than 3 authors.

     This claim is proven by Theorem 4.3 in section 4.2 given Lemma A.6 and A.7.

- Optimizing individual fairness is NP-hard, and group fairness can be solved using LP solver.

    This claim is proven by Theorem 5.11 in section 5.2.

- The proposed method is effective compared to current rejection policy.

    This claim is supported by Example 6.1. Though the case study is clearly presented, it is a hypothetical scenario and does not have real-world data to support it. I understand that collecting such data is very difficult due to the anonymity of submissions and privacy protection, but it is challenging to assess the severity of the fairness issue argued in this paper and whether the proposed method should be adopted for modifications in current AI conference.

**Essential References Not Discussed:**

No.

**Experimental Designs Or Analyses:**

This paper does not have an experimental section.

**Methods And Evaluation Criteria:**

This paper does not provide any experiment, so the evaluation criteria (no benchmark datasets, no baselines) may feel less convincing in terms of practical validation. While the theoretical framework presented is well-grounded and make sense for addressing the problem, the lack of empirical results weakens the paper's ability to demonstrate the real-world effectiveness and applicability.

**Other Comments Or Suggestions:**

n fairness literature, group fairness is more about ensuring some parity using statistical measures across different demographic groups, while individual fairness is more about treating similar individuals similarly. From the definitions in this paper, individual fairness is used to measure the worst-case cost for an author, so it might be better to name it directly as “worst-case fairness.” For group fairness, since it uses the average amount, it might be more appropriate to call it “average fairness” to avoid confusion with the conventional definitions used in fairness literature.

**Other Strengths And Weaknesses:**

Strengths:
- The novelty is strong. This paper addresses a timely fairness concern in the AI research community that has not been rigorously studied before.
- The proofs are rigorous, and the theoretical discussion of the problem is solid.
- I believe the paper has potential for practical impact.

Weakness:
- No empirical analysis, either from the perspective of the severity of the problem or the effectiveness of the method. For example, how can the actual bias mitigation be quantified after applying such a new policy?
- The NP-hardness of individual fairness optimization makes this framework impractical for real-world applications.

**Questions For Authors:**

I understand that collecting such data is challenging, but would it be possible to validate the method, even on a small scale, using real-world datasets?

**Relation To Broader Scientific Literature:**

The key contribution is related to fairness in machine learning in the context of AI conference paper submissions. To the best of my knowledge, although fairness in machine learning has been discussed in multiple real-world applications, such as NLP tasks or vision tasks, I have not seen any prior research addressing this particular situation. Therefore, from this perspective, this paper is quite novel.

**Theoretical Claims:**

Yes, I have checked the proofs in the appendix for the theoretical claims, and they look correct to me.

---

> ### Author Rebuttal · Authors · 2025-04-01
>
> We sincerely thank the reviewer for recognizing the novelty, mathematical rigor, and potential social impact of our work. We appreciate the constructive feedback and address the concerns as follows:
>
> ### **Weakness 1: The Severity of the Problem**
>
> We acknowledge that direct evaluation on real conference desk‐rejection data is challenging due to data access restrictions. Although a direct evaluation is infeasible at this stage, several indirect observations support the issue’s significance.
>
> 1) First, submission numbers in conferences have surged. For instance, NeurIPS increased from 6,743 submissions in 2019 to 15,671 in 2024, and ICML from 3,424 to 9,653 in the same period.
> 2) Moreover, Table 1 of our paper shows that many major conferences (e.g., CVPR, ICCV, AAAI, IJCAI, KDD, WSDM) have adopted submission-limit-based desk-rejection policies.
> 3) Besides, we conducted a new analysis on the ICLR 2023 data, collecting 12,451 authors and 3,793 papers using the OpenReview API. Under an 8-paper limit (as applied by IJCAI 2021–2025), we found that 506 papers (13.3%) had authors exceeding the limit, affecting 2,114 authors (17.0%). This new evidence further strengthens the urgency of addressing fairness issues in current policies.
>
> ### **Weakness 2 & Question 1: Effectiveness of the Method and Real-World Validation**
>
> Although direct testing on real backend desk-rejection data is difficult, we have empirically evaluated our approach using **public ICLR (2021–2023) data from the OpenReview API**. We compared the current desk-rejection method (Algorithm 2, page 8), which rejects all papers with non-compliant authors based solely on submission order, with our proposed group fairness optimization method (Algorithm 1, page 7), which prioritizes rejecting papers from senior researchers with many submissions to better protect junior researchers. Using the Python PuLP library and setting submission limits $x=[4,6,8,10,12,14]$, we obtained the following results:
>
> **Table R.1: ICLR ’21 Results**
>
> | Method / Limit $x$|4|6| 8 |10|12|14|
> |---------------------------------|-------|-------|-------|-------|-------|-------|
> | Cur. Desk. Rej.| 0.112 | 0.059 | 0.033 | 0.021 | 0.013 | 0.009 |
> | Fair Desk. Rej. | 0.074 | 0.035 | 0.018 | 0.011 | 0.006 | 0.004 |
>
> **Table R.2: ICLR ’22 Results**
>
> |Method / Submission Limit $x$ |4 |6 | 8     | 10    | 12    | 14    |
> |---------------------------------|-------|-------|-------|-------|-------|-------|
> |Cur. Desk. Rej. | 0.112 | 0.059 | 0.035 | 0.023 | 0.013 | 0.007 |
> |Fair Desk. Rej. | 0.073 | 0.036 | 0.019 | 0.010 | 0.005 | 0.002 |
>
> **Table R.3: ICLR ’23 Results**
>
> | Method / Limit $x$ | 4     | 6     | 8     | 10    | 12    | 14    |
> |---------------------------------|-------|-------|-------|-------|-------|-------|
> | Cur. Desk. Rej. | 0.115 | 0.056 | 0.031 | 0.022 | 0.015 | 0.009 |
> | Fair Desk. Rej. | 0.074 | 0.033 | 0.018 | 0.011 | 0.007 | 0.004 |
>
> The results in these tables use the group fairness metric (Definition 5.5), representing the average cost function (Definition 5.2) across all authors. For example, a value of 0.02 indicates that, on average, an author has 2% of their papers rejected. Since a lower fairness metric indicates a stronger fairness guarantee, we can conclude that our method consistently achieves a significant cost reduction compared to the conventional approach.
>
> ### **Weakness 3: NP-Hardness of Individual Fairness**
>
> We agree that it may not be practical to directly solve the mixed programming problem in Definition 5.7 and Definition 5.9. First, we would like to clarify that proving the hardness for the individual fairness could be considered a strong technical contribution of our work. By establishing that under submission-limit-based desk-rejection the individual fairness objective may be impossible to optimally optimize, we highlight critical fairness concerns inherent in the desk-rejection mechanism.
>
> In real-world scenarios, however, we can relax the discrete constraints on $r$ and employ a softmax transformation to make the infinity norm continuous. Concretely, we consider the alternative objective:
> $$\min_{r \in [0,1]^m}  \mathsf{Softmax}(1_n - D^{-1}Wr) \quad \text{s.t.} \quad (Wr)/x \leq 1_n.$$
> This formulation allows for smooth optimization with standard convex solvers. We acknowledge that the minimizer of this relaxed objective is not guaranteed to be the minimizer of the original problem (Definitions 5.7 and 5.9), and we leave an empirical evaluation of this approximation to future work.
>
> Another feasible approach is to use existing mixed-integer programming solvers. Considering some instances that are not worst-case and in moderate-sized conferences, this alternative may also yield acceptable solutions in practice.
>
> ### **Other Comment 1: Name of Fairness Metrics**
>
> We thank the reviewer for the suggestion on the names of fairness metrics. We will adopt your advice and improve the naming in the next version of our paper.

---

> > ### Comment · Reviewer_UGt2 · 2025-04-02
> >
> > Thank you for addressing my concerns. The supplementary experiments are helpful and, in my view, should be included in the revised version of the paper, as empirical validation is important. Also, could you please add the discussion of NP-hardness in the revised version?

---

> > > ### Author Response · Authors · 2025-04-02
> > >
> > > We are glad to hear that our response addressed your concerns! Thank you for your positive feedback on the supplementary experiments. We agree that empirical validation is important and will include these results in the revised version of the paper. We will also add a discussion of the NP-hardness, as suggested. We sincerely appreciate your thoughtful comments and valuable suggestions.

---

### Decision · Program_Chairs · 2025-05-01

**Decision:**

Accept (poster)

**Comment:**

The reviewers all generally agreed that this paper studies a timely and novel question in the design of policies for large academic conferences (such as ICML or NeurIPS), and provide non-trivial theoretical insights on this question. All reviewers were in favor of acceptance. The authors are encouraged to incorporate their responses and address some of the comments of the reviewers in the revision (e.g. the potential equilibrium effects pointed out by reviewer q5TE).